# Combined Newborn Screening Allows Comprehensive Identification also of Attenuated Phenotypes for Methylmalonic Acidurias and Homocystinuria

**DOI:** 10.3390/nu15153355

**Published:** 2023-07-28

**Authors:** Elena Schnabel, Stefan Kölker, Florian Gleich, Patrik Feyh, Friederike Hörster, Dorothea Haas, Junmin Fang-Hoffmann, Marina Morath, Gwendolyn Gramer, Wulf Röschinger, Sven F. Garbade, Georg F. Hoffmann, Jürgen G. Okun, Ulrike Mütze

**Affiliations:** 1Division of Child Neurology and Metabolic Medicine, Dietmar Hopp Metabolic Center, Center for Child and Adolescent Medicine, University Hospital Heidelberg, 69120 Heidelberg, Germany; elena.schnabel@med.uni-heidelberg.de (E.S.); juergenguenther.okun@med.uni-heidelberg.de (J.G.O.); 2Department for Inborn Metabolic Diseases, University Children’s Hospital, University Medical Center Hamburg-Eppendorf, 20246 Hamburg, Germany; 3Labor Becker MVZ GbR, Newborn Screening Unit, 81671 Munich, Germany

**Keywords:** neonatal screening, inherited metabolic disorders, vitamin B_12_ deficiency, cobalamin deficiency, propionic acidemia, remethylation disorders

## Abstract

Newborn screening (NBS) programs are effective measures of secondary prevention and have been successively extended. We aimed to evaluate NBS for methylmalonic acidurias, propionic acidemia, homocystinuria, remethylation disorders and neonatal vitamin B_12_ deficiency, and report on the identification of cofactor-responsive disease variants. This evaluation of the previously established combined multiple-tier NBS algorithm is part of the prospective pilot study “NGS2025” from August 2016 to September 2022. In 548,707 newborns, the combined algorithm was applied and led to positive NBS results in 458 of them. Overall, 166 newborns (prevalence 1: 3305) were confirmed (positive predictive value: 0.36); specifically, methylmalonic acidurias (N = 5), propionic acidemia (N = 4), remethylation disorders (N = 4), cystathionine beta-synthase (CBS) deficiency (N = 1) and neonatal vitamin B_12_ deficiency (N = 153). The majority of the identified newborns were asymptomatic at the time of the first NBS report (total: 161/166, inherited metabolic diseases: 9/14, vitamin B_12_ deficiency: 153/153). Three individuals were cofactor-responsive (methylmalonic acidurias: 2, CBS deficiency: 1), and could be treated by vitamin B_12_, vitamin B_6_ respectively, only. In conclusion, the combined NBS algorithm is technically feasible, allows the identification of attenuated and severe disease courses and can be considered to be evaluated for inclusion in national NBS panels.

## 1. Introduction

Newborn screening (NBS) has been widely established and significantly extended by tandem mass spectrometry (MS/MS) [1,2]. The decision of disease inclusion into NBS programs is generally based on the principles of screening defined by Wilson and Jungner [3]; however, national NBS programs differ significantly in terms of disease panels and implementation pathways for new conditions.

Candidate metabolic disorders for NBS include methylmalonic acidurias, propionic acidemia, as well as homocystinurias, remethylation disorders and neonatal vitamin B_12_ deficiency. Screening for methylmalonic aciduria and propionic acidemia based on a single-tier analysis of propionyl carnitine (C3) implies high false positive rates, pointing to the need for a second-tier strategy [4,5]. In contrast, for homocystinurias NBS based on homocysteine (Hcy) as a primary marker showed high sensitivity and specificity [6] but would not be applicable in a high-throughput screening for western countries [7,8]. Therefore, combined multiple-tier algorithms have been implemented, and were recently introduced to several NBS pilot panels [4,7,9,10,11,12,13,14,15,16].

So far, NBS has been reported to be beneficial for vitamin B_12_ deficiency [17] and homocystinurias, including remethylation disorders [18]; however, the latter diseases are ultra-rare and thus, NBS for single diseases is not cost-effective [8]. Additionally, it was previously assumed that cofactor-responsive disease variants, especially for homocystinuria, might be missed by NBS [19,20]. For methylmalonic acidurias and propionic acidemia, already included in some European NBS programs [21], the benefit of NBS is still discussed controversially and without consensus in recommendation [22].

The prospective NBS pilot study evaluating 28 additional conditions (“Newborn screening 2020/2025” (NGS2025)) established at Heidelberg University Hospital (UKHD) includes a combined algorithm with multiple-tier strategy for methylmalonic acidurias, propionic acidemia, homocystinurias, remethylation disorders and neonatal vitamin B_12_ deficiency [7,8,9,23,24,25].

The aim of the present study is to evaluate this combined screening algorithm with a specific focus on the case descriptions of two identified individuals with attenuated cofactor-responsive disease variants.

## 2. Materials and Methods

### 2.1. Newborn Screening

NBS in Germany is legally regulated by the national NBS directive [26] and must be offered to every newborn, but parental written consent is necessary. To date, the German national panel includes 19 conditions, 13 of which are inherited metabolic diseases (IMDs) [26]. Samples should be taken between 36 and 72 h of life on dried blood spot cards (DBS, Neonatal Screening Card; Whatman filter paper 903, GE Healthcare Europe GmbH, Freiburg, Germany), dried at room temperature, sent to one of the national NBS laboratories by mail, and analyzed within 24 h after arrival at the NBS laboratory [26].

Metabolic NBS includes analysis of amino acids and acylcarnitine profile by electrospray ionization tandem mass spectrometry (MS/MS; Micromass Ultima, Waters Xevo TQD, Waters, Milford, MA, USA), as previously described [2,24].

### 2.2. NGS2025 as Extension of the German National NBS Panel

Since 1st August 2016, the prospective NBS pilot study (NGS2025) extending the German national NBS panel by 28 additional conditions has been performed at the NBS laboratory of UKHD, Germany [27]. NGS2025 aims to evaluate the technical feasibility of population-based NBS for IMDs not included in the national NBS disease panel, and the benefit for early diagnosed and treated patients [28].

### 2.3. Combined Multiple-Tier NBS Algorithm

A combined multiple-tier NBS algorithm was implemented as part of the NGS2025 panel to screen for methylmalonic acidurias, propionic acidemia, homocystinurias, remethylation disorders and neonatal vitamin B_12_ deficiency (Figure 1) [9,23,24] using methionine (Met), Met-phenylalanine ratio (Met/Phe), C3, C3-acetylcarnitine ratio (C3/C2) as primary biomarkers (first-tiers) and Hcy and/or methylmalonic acid (MMA), 3-hydroxypropionic acid (3-OH-PA), and methylcitrate (MCA) as second- or third-tiers, determined in the same NBS sample (Figure 1). First-tier metabolites were measured by MS-MS [2]. Pathological results of first-tier parameters triggered determination of second- and third-tier parameters in DBS by a liquid chromatography-tandem mass spectrometry assay [6,8,24].

The algorithm thereby allows the identification of: Methylmalonic acidurias: methylmalonyl-CoA mutase deficiency (MUT-type methylmalonic aciduria; OMIM #251000), cobalamin (Cbl) A deficiency (CblA-type methylmalonic aciduria; OMIM #251100) and CblB deficiency (CblB-type methylmalonic aciduria; OMIM #251110); propionic acidemia (PA; OMIM #606054); cystathionine beta-synthase deficiency (CBS; OMIM #236200); isolated remethylation disorders: 5,10-methylenetetrahydrofolate reductase deficiency (MTHFR; OMIM #236250; thermolabile variant excluded), CblD-Hcy deficiency (OMIM #277410), methionine synthase reductase deficiency (CblE; OMIM #236270) and methionine synthase deficiency (CblG; OMIM #250940); combined remethylation disorders: CblC deficiency (OMIM #277400), CblD deficiency (OMIM #277410), CblF deficiency (OMIM #277380), CblJ deficiency (OMIM #614857), CblX deficiency (OMIM #309541) and transcobalamin II deficiency (TCN2; OMIM #275350); as well as neonatal vitamin B_12_ deficiency.

Diagnoses were biochemically and genetically confirmed for IMDs in accordance with existing guidelines [22,29,30], and biochemically for vitamin B_12_ deficiency [17,31].

### 2.4. Ethical Approval

The NGS2025 pilot study (German Clinical Trials Register identifier: DRKS00025324) was approved by the local ethics committee (Medical Faculty; University Heidelberg; Vote S-533/2015). Inclusion criteria were (1) regular NBS analyzed at Heidelberg NBS laboratory and (2) written informed consent from parents. Long-term outcome of confirmed individuals identified by NBS is evaluated in the Heidelberg prospective multicenter observational study (German Clinical Trials Register identifier: DRKS00013329; [28]) approved by the local ethics committee of the Medical Faculty of Heidelberg (Vote S-104/2005) and consecutively by the study sites (in Düsseldorf, Freiburg, Mainz, Reutlingen, and Ulm, Germany). Parents’ written informed consent to the outcome study was given prior to data collection.

### 2.5. Statistical Analysis 

Statistical analysis was carried out with Microsoft Excel and R language for statistical computation (Version 4.2.2). We used the Mann-Whitney test to compare age at the start of therapy between IMDs and individuals with vitamin B_12_ deficiency. We used R package “epiR” (Version 2.0.58) to estimate birth prevalences with 95% exact confidence levels. 

## 3. Results

### 3.1. Study Population

Altogether, 879,975 newborns were screened at the NBS laboratory of UKHD between 1 August 2016 and 30 September 2022 (Figure 2), accounting for approximately 18% of children born in Germany during this period [32]. 62.3 % of newborns screened at UKHD (N = 548,707) participated in the NGS2025 study (Figure 2).

### 3.2. Suspected and Confirmed Diagnoses

In 8.2% (N = 44,845) of the participating newborns, abnormal first-tier results necessitated second-tier analysis of Hcy (96.6%; N = 43,304) and/or MMA, 3-OH-PA and MCA (3.4%; N = 1541; Figure 2). In 458 newborns (1:1198 of all NGS2025 participants; 1:98 of those with second-tier analysis), NBS resulted in a positive NBS report and confirmatory tests (Figure 2). In 166 newborns (1: 3305 screened newborns), one of the conditions of the combined algorithm was confirmed (Table 1, Figure 2), with an overall positive predictive value of 0.36. The most frequent diagnosis was neonatal vitamin B_12_ deficiency (N = 153). Additionally, we identified individuals with propionic acidemia (N = 4), methylmalonic acidurias (N = 4 MUT-type; N = 1 CblA-type), isolated remethylation disorders (N = 3 MTHFR deficiency), combined remethylation disorder (N = 1 CblC deficiency) and CBS deficiency (N = 1) (Table 1, Figure 2 and Figure 3). These IMDs were biochemically and subsequently also genetically confirmed. Coincidentally, the CBS deficiency patient had neonatal vitamin B_12_ deficiency, too. Of note, in 42 newborns (25.3% of the confirmed), the confirmed diagnosis differed from the initially suspected one (Table 1, Figure 3, Appendix A).

As far as we are aware of, we did not miss any individual with an IMD included in the pilot panel, while six individuals developed weeks to months after the negative NBS report an infantile vitamin B_12_ deficiency, and were diagnosed following the onset of symptoms [8]. During the study period, one child with methylmalonic aciduria whose parents had not consented to the pilot study developed a metabolic decompensation with severe metabolic acidosis, hyperammonemia (290 µmol/l), hypoglycemia and seizures at the age of 10 months.

### 3.3. Time to Treatment

Overall, first NBS results were reported at a median (range) age of 8 (0–52) days (Table 2). In individuals with IMDs, treatment started immediately after the first NBS report [7.5 (0–59) days] and prior to the confirmation of the suspected diagnosis to prevent the manifestation of a potentially life-threatening metabolic decompensation and irreversible damage in the meantime. For individuals with suspected vitamin B_12_ deficiency, treatment started straight after confirmation of the diagnosis [31 (6–157) days; *p* < 0.0001]. Of note, all newborns with vitamin B_12_ deficiency (99.3%) except one with delayed start of treatment and 64.3% (9/14) of the individuals with IMDs were asymptomatic at the start of treatment (Table 2). In two newborns with known index patients for propionic acidemia, NBS was taken directly after birth and treatment was already started prior to the NBS report (PA-3, PA-4; Table 2). In one patient with MTHFR deficiency (MTHFR-3; Table 2), the report of the positive NBS result was delayed until age 52 days due to IT-technical complications in second-tier evaluation.

Unexpectedly, the combined algorithm also identified attenuated and cofactor-responsive variants: two individuals with vitamin B_12_-responsive methylmalonic acidurias and one individual with vitamin B_6_-responsive CBS deficiency. In the following, two of these cases are presented in more detail.

### 3.4. Case A: Vitamin B_6_-Responsive CBS Deficiency

Patient A is the first child of non-consanguineous parents with European origin, born at term via secondary C-section after uncomplicated pregnancy, except for a mild maternal SARS-CoV-2 infection. Vitamin supplementation but no other treatment in pregnancy was reported. Birth weight (3665 g), length (55 cm), head circumference (34 cm) and postnatal adaption were normal. No neonatal complications were reported. The NBS sample was taken at 41 h of life and arrived 100 h later at the NBS laboratory. A slight increase of the Met/Phe ratio above the upper cut-off triggered second-tier analysis of Hcy, which was also slightly increased and, therefore, homocystinuria was suspected (Table 3). A second DBS card was requested at day 9 and taken at day 12, but accidentally sent to another German NBS center. This NBS center also performed an NBS pilot study including homocystinurias and neonatal vitamin B_12_ deficiency. The initial suspicion was confirmed (Table 3). In an out-patient setting, confirmatory diagnostics were performed, vitamin B_12_ deficiency subsequently confirmed and oral treatment with vitamin B_12_ (500 µg/d for three days, following 100 µg/d oral) initiated at age 27 days (Table 3). Metabolic control after six weeks of treatment revealed normalization of the vitamin B_12_ status, but increased Hcy and Met concentrations in plasma, pathognomonic for CBS deficiency. Due to holiday leave of the family, only folic acid substitution was re-started in a higher dose immediately, and three weeks later the patient was eventually admitted to hospital for further confirmatory tests as well as the test for vitamin B_6_-responsiveness (according to [30]; started at age 102 days). On oral vitamin B_6_ supplementation (100 mg/day), Hcy decreased and remained constantly below 50 µmol/l, while Met normalized (Figure 4A). According to the classification of Kozich and colleagues [19], the child was classified as a full vitamin B_6_-responder. The patient receives 50 mg vitamin B_6_ per day, while Hcy remains below 50 µmol/l, and vitamin B_6_ is gradually adjusted to the minimally effective dosage.

Genetic testing confirmed CBS deficiency: Heterozygosity of the following variants in *CBS* gene c.828 + 1G > A; p.? (pathogenic; previously described in individuals with CBS deficiency) and c.784A > G; pThr262Ala (likely pathogenic; variant not described yet).

Until the latest follow-up at the age of 8 months, the child is asymptomatic and developing age appropriately.

### 3.5. Case B: Vitamin B_12_-Responsive Methylmalonic Aciduria, CblA-Type

Patient B is the ninth child of consanguineous parents from the Middle East born at term via primary C-section after uncomplicated pregnancy without known maternal medication or supplementation. Birth weight (3650 g), length (52 cm), head circumference (34 cm) and postnatal adaption were normal. The NBS sample was taken at age 40 h and arrived 96 h later at the NBS laboratory. Increased C3 and C3/C2 ratio resulted in the second-tier analysis for MMA, 3-OH-PA and MCA, and, already before receiving the second-tier parameters, NBS was reported as suspicious for propionic acidemia. MMA, 3-OH-PA and MCA were all increased, while Hcy was normal (Table 4), pointing to a methylmalonic aciduria. A second DBS sample was requested and obtained at age 6 days, confirming the suspected diagnosis. Emergency treatment (protein restriction, glucose infusion and carnitine supplementation) was started at day 7. Subsequently, after biochemical confirmation of a high MMA excretion in urine, vitamin B_12_-responsiveness was tested (according to [34]). After administration of 1 mg hydroxycobalamin intramuscular for five consecutive days, MMA significantly decreased to near-normal concentrations, confirming vitamin B_12_-responsiveness (Figure 4B). Protein restriction was stepwise reduced and, finally, the child was discharged with normal protein intake and oral carnitine supplementation. During the whole inpatient stay, the patient remained clinically asymptomatic and did not provide any evidence for (impending) metabolic decompensation.

Genetic testing confirmed a CblA-type methylmalonic aciduria with homozygous pathogenic variants (c.586C > T; p.Arg196 *) in the *MMAA* gene.

Until the latest follow-up at the age of 7 months, the child remained asymptomatic with no signs of metabolic decompensation and showed an age appropriate development. In the eight siblings, methylmalonic aciduria was excluded.

## 4. Discussion

We confirm that the combined NBS algorithm performed at UKHD since 2016 is technically feasible and allows reliable identification of methylmalonic acidurias, propionic acidemia, homocystinurias, remethylation disorders and neonatal vitamin B_12_ deficiency, including cofactor-responsive attenuated disease variants. The high participation rate reflects high acceptance by parents.

Overall, birth prevalence of all diseases included in the combined algorithm was 1:3305 (95% CI: 1:3297–3314), comparable to the existing metabolic panel of the national NBS [28].

The prevalence for the single diseases of the combined algorithm differed widely, with vitamin B_12_ deficiency being the most frequent. In 25%, however, as shown by the present work, the initial suspected diagnosis did not correspond to the confirmed diagnosis, indicating the high overlap of metabolic patterns of the diseases. This is in line with a previous work showing first-tier parameters indicating vitamin B_12_ deficiency to be not completely homogeneous [9]. While in 82% remethylation pathway metabolites (Met, Met/Phe) were most prominent, C3-based markers (C3, C3/C2) were leading in 9%, and another 9% presented with a combined pattern of both [9]. In general, MMA and Hcy in plasma or dried blood are less elevated in vitamin B_12_ deficiency [31] than in methylmalonic acidurias, propionic acidemia, homocystinurias and remethylation disorders [22,29,30]. However, IMDs present as a continuous spectrum regarding enzyme activity and biochemical specificities. Especially, attenuated variants are not always distinguishable from other IMDs or vitamin B_12_ deficiency when only the first metabolic pattern is taken into account. However, screening with the combined algorithm allowed identifying affected individuals, although their first pattern was ambiguous. With this capacity, this NBS strategy is a strong tool with a high PPV (Table 1), and strengthens the suggestion to use a combined screening for these conditions [11]. Furthermore, the focus on selected conditions would always identify others out of these metabolic pathways too.

The combined screening algorithm is based on a multiple-tier strategy (Figure 1), which could be shown to be suitable for high throughput NBS [8,24]. Analysis of the second- and third-tier parameters requires more time than the current MS/MS-based metabolic screening. However, 64% of the newborns with IMDs and 100% of the newborns with neonatal vitamin B_12_ deficiency were still asymptomatic at the time of the positive NBS report. The remaining five patients showed mild symptoms and could clearly benefit from early diagnosis and treatment. We assume that identification by NBS might have prevented a (neonatal) decompensation in some of the individuals.

However, the report of suspicious screening was delayed in one child due to IT-technical complications in second-tier evaluation. From the technical point of view, this default is of special interest, as the algorithm is hampered by the necessary high Hcy second-tier rate (about 8% of all samples) and, therefore, needs special attention [8].

A general limitation of NBS programs is the identification of attenuated disease variants that were underestimated or even unknown in the pre-screening era (e.g., attenuated vs. classical isovaleric acidemia, wide biochemical spectrum of medium-chain acyl-CoA dehydrogenase deficiency [39,40]), which might lead to harm and overtreatment of the identified individuals [39]. For other diagnoses like urea cycle disorders, especially attenuated variants benefit from early detection via NBS [41], while NBS might be too late for severe variants.

For the IMDs screened by the presented algorithm, no rise of prevalence was observed, although it also identified the attenuated cofactor-responsive disease variants. This has to be mentioned as an explicit advantage of this screening design. Hitherto, vitamin B_6_-responsive CBS deficiency has been thought to be missed by NBS [19].

We could firstly report a patient with vitamin B_6_-responsive CBS deficiency identified by NBS. Early identification by NBS is very likely to prevent later disease-specific symptoms and complications in individuals with CBS deficiency [19,30]. Ophthalmologic symptoms (e. g. lens luxation) and thromboembolic events are mostly preventable under guideline-compliant therapy, and neurologic symptoms and developmental disability are at least significantly mitigated, which is especially true for the attenuated cofactor-responsive disease variants [19,20,30].

The second presented child was diagnosed with a full vitamin B_12_-responsive CblA-type methylmalonic aciduria. Isolated methylmalonic acidurias include an etiologically heterogeneous group of diseases that are biochemically characterized by an increased concentration of MMA in blood and other body fluids without increased concentration of Hcy. Specifically, it refers to a complete or partial deficiency of methylmalonyl-CoA mutase (MUT-type methylmalonic aciduria, MMUT gene) or congenital defects in the metabolism of the cofactor adenosyl cobalamin (CblA-type methylmalonic aciduria, MMAA gene; CblB-type methylmalonic aciduria, MMAB gene). Individuals with isolated methylmalonic acidurias present with acute, potentially life-threatening (neonatal) metabolic decompensations, which can lead to irreversible damage of the central nervous system. In the long-term course, affected individuals might develop chronic progressive multi-organ involvement, in particular, neurological (epilepsy, developmental and movement disorders) and renal manifestations (chronic renal failure), and less frequently, pancreatitis, optic atrophy and cardiomyopathy [22,34]. However, the different disease groups differ in severity of the clinical phenotype [22,34,42,43,44]. A response to hydroxycobalamin supplementation, which is frequently found in CblA, but rarely in CblB and not in MUT-type, is prognostically favorable [22,43,44,45]. Cobalamin-responsive (cbl+) patients have a longer life expectancy and develop neurological and renal organ complications less frequently and later than nonresponsive (cbl-) patients [45]. Thus, hydroxycobalamin response has to be tested in every diagnosed methylmalonic aciduria according to a standardized protocol [34].

NBS for methylmalonic acidurias aims to diagnose the disease before onset of an acute metabolic decompensation. Investigating the potential benefit of NBS, it has been shown that on day 8 of life, 59% of cbl- and 72% of cbl+ patients are still asymptomatic and could be treated pre-symptomatically (Table 2) [46]. Accordingly, in our cohort, only two of the five patients identified with methylmalonic aciduria were symptomatic, and these two showed only mild symptoms. Concerning long-term outcome, identification by NBS was associated with a moderate risk reduction for long-term damage from the first metabolic crisis [47], and also a reduction of the risk to develop a motor developmental delay or a movement disorder for vitamin B_12_-nonresponsive individuals with methylmalonic aciduria [46]. Long-term follow-up data of our cohort are currently being collected.

## 5. Conclusions

Our experience has shown that the combined screening algorithm is technically feasible as a high-throughput analysis in daily routine and, therefore, allows NBS for methylmalonic acidurias, propionic acidemia, homocystinurias, remethylation disorders and neonatal vitamin B_12_ deficiency. The low false positive rates and the respective high PPV (about 0.3) indicate an acceptable risk of possible harm for unaffected newborns and their families. In addition, the previously known birth prevalence was not substantially increased. Considering the necessity of an additionally required consent, the participation rate of 62% of all screened newborns reflects a high acceptance by parents. As shown, not only severe, but also attenuated disease forms could be identified, which formerly were thought to be missed by NBS. In conclusion, the combined screening algorithm can be considered a candidate to be evaluated for inclusion in national NBS panels.

## Figures and Tables

**Figure 1 nutrients-15-03355-f001:**
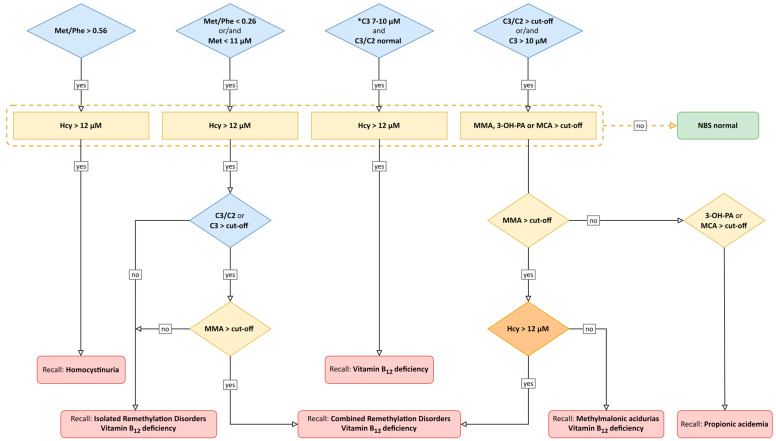
Combined Newborn Screening Algorithm. Figure adapted according to [9]. The diagnostic scheme for the combined algorithm used within the NGS2025 pilot panel allows diagnosis of methylmalonic acidurias, propionic acidemia, homocystinurias, isolated and combined remethylation disorders and neonatal vitamin B_12_ deficiency. First-tier parameters (blue): Methionine (Met), met-phenylalanine ratio (Met/Phe), propionyl carnitine (C3) and C3-acetylcarnitine ratio (C3/C2); Second-tier (yellow) and third-tier (orange) parameters: Homocysteine (Hcy), methylmalonic acid (MMA), 3-hydroxypropionic acid (3-OH-PA) and methylcitrate (MCA). Cut-offs: Met < 11 µmol/l (Perc 5), Met/Phe < 0.26 (Perc 5) or > 0.56 (Perc 89), C3 5.5 µmol/l (Perc 99–99.5), C3/C2 0.22 (Perc 99.5), MMA 2.35 µmol/l (Perc 99.9), 3-OH-PA 43.32 µmol/l (Perc 99.9), MCA 0.34 µmol/l (Perc 99.9). * this pathway was introduced in 2020 as adjustment to the results of [9]. Figure was created with draw.io (https://drawio-app.com/, accessed on 21 April 2023).

**Figure 2 nutrients-15-03355-f002:**
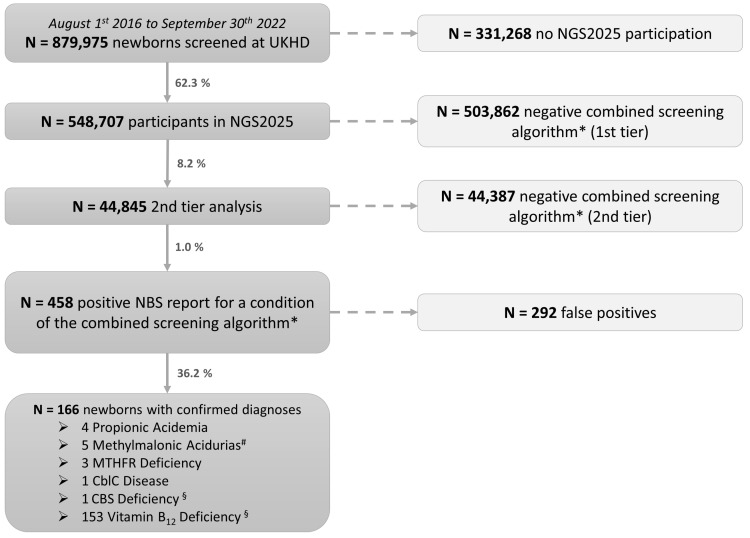
Screening and study sample. Altogether, 879,975 newborns were screened at UKHD during the study interval, and 548,707 of them participated in the NGS2025 pilot study, which includes 44,845 2nd-tier analyses and 458 positive NBS reports. A metabolic disease of the combined algorithm (*) including methylmalonic acidurias, propionic acidemia, homocystinurias, isolated and combined remethylation disorders and neonatal vitamin B_12_ deficiency was confirmed for 166 newborns. ^#^ including four patients with MUT-type and one patient with CblA-type methylmalonic aciduria. ^§^ including one patient with both CBS and neonatal vitamin B_12_ deficiency.

**Figure 3 nutrients-15-03355-f003:**
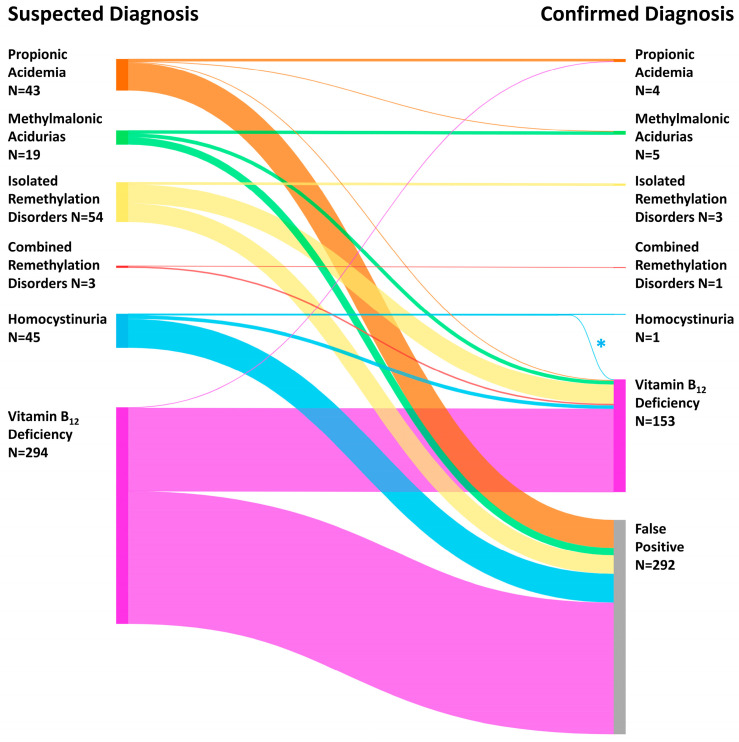
Suspected versus Confirmed Diseases. 166 newborns were diagnosed with a metabolic disease; however, for 42 individuals, the confirmed diagnosis was different to the initial NBS report (Table 1, Appendix A). One patient was diagnosed with CBS and vitamin B_12_ deficiency (*) and is counted twice. Major changes occurred for suspected organic acidurias (methylmalonic acidurias, propionic acidemia) to false positives and for suspected homocystinurias and remethylation disorders to vitamin B_12_ deficiency. Exact numbers are shown in Table 1. Figure was created with SankeyMATIC (https://sankeymatic.com/build/, accessed on 9 February 2023) with flows corresponding in proportion to actual patient numbers.

**Figure 4 nutrients-15-03355-f004:**
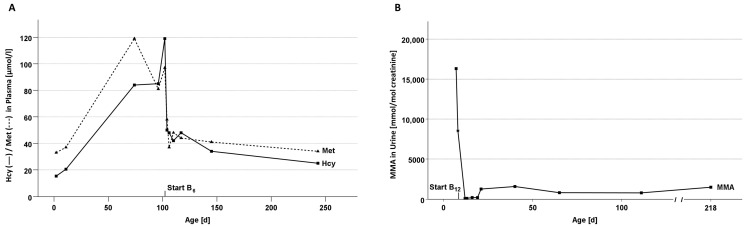
(**A**): Plasma Concentration of Homocysteine (Hcy) and Methionine (Met) with and without Vitamin B_6_ treatment in Case A. Vitamin B_6_-responsive CBS deficiency; medication: pyridoxine; 100 mg/d oral. (**B**): Urine Concentration of Methylmalonic Acid (MMA) with and without Vitamin B_12_ treatment in Case B. Vitamin B_12_-responsive methylmalonic aciduria; medication: hydroxycobalamin; 1 mg/d i. m. * median of four measurements taken on day 8 (8521 mmol/mol creatinine). Figure was created with IBM SPSS Statistics 28.0.0.0.

**Table 1 nutrients-15-03355-t001:** Suspected and confirmed diagnoses of the combined screening algorithm. Total screened newborns (NGS2025): N = 548,707. B_12_D: Vitamin B_12_ Deficiency; MMA: Methylmalonic Acidurias; PA: Propionic Acidemia. Diagnoses were confirmed biochemically and for all IMDs also genetically. Changes between initially suspected diagnosis and thereafter confirmed diagnosis are depicted in Figure 3 and listed in Appendix A. * including one patient with CBS deficiency and vitamin B_12_ deficiency who is counted twice; ^#^ includes 170 false positive, nine lost to follow-ups and one patient who died for other reasons prior to confirmation. ^§^ estimated incidence (acquired disease).

	Suspected Diagnosis (N)	Results of Confirmatory Tests			Confirmed Diagnosis (N)	Positive Predictive Value (PPV)		Estimated Birth Prevalences [1:X]	
		Confirmation of the Initially Suspected Diagnosis	Confirmation of Another Diagnosis of the Algorithm	False Positives		For the Initially Suspected Diagnosis	For Any Diagnosis of the Algorithm	Study Cohort [95% CI]	Reports from Literature
Propionic Acidemia (PA)	43	3	2 (1× B_12_D, 1× MMA)	38	4	0.07	0.12	137,177 [136,814–137,540]	5000 (Saudi Arabia) [33]–150,000 [34,35]
Methylmalonic Acidurias (MMA)	19	4	5 (B_12_D)	10	5	0.21	0.47	109,741 [109,451–110,032]	50,000–200,000 [10,34,35]
Isolated Remethylation Disorders	54	3	26 (B_12_D)	25	3	0.06	0.54	182,902 [182,419–183,387]	No valid data [20]
Combined Remethylation Disorders	3	1	2 (B_12_D)	0	1	0.33	1	548,707 [547,256–550,161]	No valid data [20]
Homocystinuria	45	1 *	6 * (B_12_D)	39	1 *	0.02	0.13	548,707 [547,256–550,161]	1800 (Qatar) [36]–200,000 [37]–900,000 [30]
Neonatal Vitamin B_12_ Deficiency (B_12_D)	294	113	1 (PA)	180 ^#^	153 *	0.38	0.39	3586 ^§^ [3577–3596]	2000 [11]–3600 [9,17,23]–30,000 [38]
TOTAL	458	125 *	42 *	292	166	0.27	0.36	3305 [3297–3314]	

**Table 2 nutrients-15-03355-t002:** NBS process data and symptoms of confirmed individuals identified by the combined screening algorithm. * presented as case reports within the manuscript. ^#^ therapy was started directly after birth prior NBS result following a high-risk family screening due to an older index sibling. B_12_D: Vitamin B_12_ Deficiency.

Disease Name, Number (N) of Patients		Age at First NBS Report [days]	Age at Start of Therapy [days]	Age at Disease Confirmation [days]	Symptomatic at First NBS Report		Additional Information
**Propionic Acidemia**							
**N = 4**	PA-1	7	8	8	No		Suspected diagnosis: Vitamin B_12_ deficiency
	PA-2	6	6	7	Yes	Mild hyperammonemia, metabolic acidosis	
	PA-3	0	at birth ^#^	0	No		High-risk screening (sibling of PA-2)
	PA-4	3	at birth ^#^	3	No		High-risk screening, pre-natal diagnosis
**Methylmalonic Acidurias**							
**N = 5**	MMA-1	15	15	16	No		Partial vitamin B_12_-responsiveness, MUT-type
	MMA-2	8	2	3	Yes	Mild hyperammonemia	MUT-type
	MMA-3	8	8	9	Yes	Mild hyperammonemia, metabolic acidosis	MUT-type
	MMA-4	3	3	4	No		MUT-type
	MMA-5 *	6	7	7	No		Full vitamin B_12_-responsiveness, CblA-type
**Isolated Remethylation Disorders**							
**N = 3**	MTHFR-1	8	8	9	No		High-risk screening
	MTHFR-2	6	6	12	No		
	MTHFR-3	52	59	59	Yes	Muscular hypotonia	Recall delayed due to IT-technical complications
**Combined Remethylation Disorders**							
**N = 1**	CblC	8	8	8	Yes	Feeding difficulties, hypothermia	
**Homocystinuria**							
**N = 1**	CBS *	9	30	30 (B_12_D)/ 102 (CBS)	No		Full vitamin B_6_-responsiveness, additional vitamin B_12_ deficiency
**Neonatal Vitamin B_12_ Deficiency**							
**N = 153**		Median (range)			No		One patient was symptomatic (mild muscular hypotonia) at start of therapy (116 days) due to delayed confirmatory diagnostics despite tracking by NBS laboratory [17]
		8.5 (5–52)	31 (6–157)	30 (6–145)			
**TOTAL**							
**N = 166**		Median (range)			N = 161 no N = 5 yes		
		8 (0–52)	30 (0–157)	29 (0–145)			

**Table 3 nutrients-15-03355-t003:** Patient A: Vitamin B_6_-responsive CBS deficiency—Results of newborn screening and confirmatory diagnostics. * sampling was difficult; thus, volume was too little to perform amino acids in plasma. ^#^ 2nd DBS card was sent to another screening center (München) with different reference ranges. ^§^ treatment was discontinued after 2 weeks and re-started on day 74.

Age at Sampling:	41 h	12 d	27 d	74 d	96 d	102 d
		ReferenceValues UKHD						
**DBS (NBS):**							
1st tier:	Met/Phe	0.26–0.56	**0.57**	**0.94**				
Met [µmol/l]	11–3511–40 (ext ^#^)	33	37				
2nd tier:	Hcy [µmol/l]	0.1–120–15 (ext ^#^)	**15.3**	**20.5**				
**Confirmatory diagnostics:**							
**Plasma:**							
	Hcy [µmol/l]	2–14			**34**	**84**	**85**	**119**
	Met [µmol/l]	15–35			- *	**119**	**81**	**97**
	MMA [µmol/l]	0–0.26			0.25	0.24	0.22	n/a
	Vitamin B_12_ [pmol/l]	160–670			**136**	959	497	576
	Folic acid [nmol/l]	4.5–21			>45	36	>45	>45
	Holo-Transcobalamin [µmol/l]	>60			**47**	>150	n/a	n/a
**Urine:**							
	MMA [mmol/molCrea]	0–10			6.7	2.5	n/a	6.1
**Treatment (all oral):**	
Vitamin B_12_:	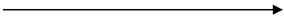
Folic acid:	 ^§^ 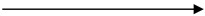
Vitamin B_6_:	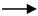

**Table 4 nutrients-15-03355-t004:** Patient B: Vitamin B_12_-responsive methylmalonic aciduria, CblA-type—Results of newborn screening and confirmatory diagnostics.

Age at Sampling:	40 h	6 d	8 d
		ReferenceValues UKHD			
**DBS (NBS):**				
1st tier:	C3 [µmol/l]	0−5.5	**16.3**	**9.1**	**53.8**
C3/2	0−0.22	**0.38**	**0.39**	**1.4**
2nd tier:	MMA [µmol/l]	0−2.35	**466.3**		
3-OH-PA [µmol/l]	0−77.5	**109.5**		
	MCA [µmol/l]	0−0.34	**7.8**		
	Hcy [µmol/l]	0.1−12	9.4		
**Confirmatory diagnostics:**				
**Plasma:**				
	Ammonium [µmol/l]	12−53			27
**Urine:**				
	MMA [mmol/mol creatinine]	0−18			**5767**
	MCA [mmol/mol creatinine]	0−9			**68**
	MMA-stable isotope quantification [mmol/mol creatinine]	0−10			**8363**
**Treatment:**				
Carnitine:	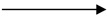
Vitamin B_12_:	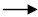

## Data Availability

De-identified individual participant data will not be made available.

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
