# Peer review of "Combined Newborn Screening Allows Comprehensive Identification also of Attenuated Phenotypes for Methylmalonic Acidurias and Homocystinuria"

_nutrients, 2023, doi:10.3390/nu15153355_

Round 1
Reviewer 1 Report
The manuscript "Combined Newborn Screening Allows Comprehensive Identification also of Attenuated Phenotypes for Methylmalonic Acidurias and Homocystinuria" addresses an important issue: evaluating the effectiveness and feasibility of newborn screening (NBS) for a range of metabolic disorders, including methylmalonic acidurias, propionic acidemia, homocystinuria, remethylation disorders, and neonatal vitamin B12 deficiency. The authors evaluated the NBS program's performance for the aforementioned disorders and report on the identification of cofactor-responsive disease variants.
The objectives were clearly stated and explained in the manuscript, however the experimental strategy raises some major concerns and so the experimental information from which the conclusions were drawn. The manuscript is overall well written and has good organization with minor English language and style spell check required. The authors have done a great job on analyzing the experimental data and on discussing the results and their limitations, considering always different alternative explanations/considerations for interpreting the results.
The paper is interesting but there is a need for more experimental detail in order to critically review the data. Specifically, they should provide information for the following questions and comments:
Major points:
1. The authors should include more recent update on this topic and compare how this study further advances the current knowledge in the “Introduction section”.
2. Unify the style of the references in the References Section and add DOI in the cases it is possible. And use the same reference and citation (follow MDPI’s guidelines) style in the main text.
3. The Methods section in the study should be more accurately described for each technique used in the Materials & Methods Section.
4. The Conclusion Section should be more thoroughly described.
Minor points:
1. The resolution and quality of some Figures is low, the authors should provide higher quality Figures specially for Figure 1.
Reviewer 2 Report
You should be congratulated on a pilot involving over one half million newborns which clearly demonstrates that the multiple tier algorithm for C3 -related disorders is feasible and can detect cofactor-responsive disease, which much more often than not has been missed in more conventional screening programs.
While this reviewer would have appreciated seeing a comparison of your results to the pattern recognition that could have been achieved through CLIR (R4S collaborative project, Mayo Clinic), I appreciate that your algorithm and its implementation is essentially stand-alone and requires minimal additional technical personnel or data entry. Such a comparison was not a goal of the NGS2025 study.
There are very few minor suggestions to the English language utilization as outlined below:
Line 108: Insert "to’ between prior and data
Line 356: Change “NBS on methylmalonic c” to “NBS of methylmalonic”
Round 2
Reviewer 1 Report
The authors have answered the proposed comments and suggestions and thus I recommend this article to be accepted.
Author Response
Thank you for your valuable advice.